# Image Classification Method Based on Multi-Agent Reinforcement Learning for Defects Detection for Casting

**DOI:** 10.3390/s22145143

**Published:** 2022-07-08

**Authors:** Chaoyue Liu, Yulai Zhang, Sijia Mao

**Affiliations:** School of Information Technology and Electronics Engineering, Zhejiang University of Science and Technology, Hangzhou 310023, China; 222008855017@zust.edu.cn (C.L.); 222008855019@zust.edu.cn (S.M.)

**Keywords:** multi-agent reinforcement learning, casting image classification, casting defects detection

## Abstract

A casting image classification method based on multi-agent reinforcement learning is proposed in this paper to solve the problem of casting defects detection. To reduce the detection time, each agent observes only a small part of the image and can move freely on the image to judge the result together. In the proposed method, the convolutional neural network is used to extract the local observation features, and the hidden state of the gated recurrent unit is used for message transmission between different agents. Each agent acts in a decentralized manner based on its own observations. All agents work together to determine the image type and update the parameters of the models by the stochastic gradient descent method. The new method maintains high accuracy. Meanwhile, the computational time can be significantly reduced to only one fifth of that of the GhostNet.

## 1. Introduction

Casting defects detection is an essential problem in the machinery industry. High quality castings are very important for automobiles, engineering equipments and other products. It is necessary to find and deal with these defective castings in time. Otherwise, it will harm the quality of downstream industrial products [1]. With the development of computer vision technology, defects detection methods based on casting image processing have been proposed and applied in the recent years [2,3,4,5]. These progresses are made upon various excellent network structures, such as AlexNet [6], VGG [7], and ResNet [8]. In the past decade, in order to improve the classification accuracy, the models are designed to be more and more complex, which leads to an increasingly large number of parameters and an exponential increase in the computational burdens, especially for large images [9,10]. The image size of the castings is also large, but usually the defects are only located in a small number of the pixels, which means large numbers of the pixels in the image are useless for the defects detection task, but they still consume equivalent computational resources. So, the reinforcement learning method with multiple agents are used in this paper to find the most informative sub-images.

Image classification based on reinforcement learning has been a research hot spot in recent years. In the work of Ref. [11], the policy gradient method [12], which is a very basic reinforcement learning algorithm, is used to optimize the convolutional neural network model for image classification. In the work of Ref. [13], the Faster R-CNN model [14] is used to implement anomaly detection while the Deep Q-learning algorithm [15] is used to classify anomalies in the images. In the above works, the reinforcement learning algorithms are used as the optimizer on all the pixels of an image or a sub-image derived by some other methods, so single agent algorithms are natural choices in such a situation. On the contrary, reinforcement learning algorithms can also be used as an effective tool to find the key region in the image. In Ref. [16], Deep Q-learning algorithm is used to select a vital area of the image in a vehicle classification task. In Ref. [17], a multi-agent algorithm PixelRL, which is based on the A3C [18] algorithm, is proposed. In PixelRL, each pixel has been assigned an agent so the number of agents is equivalent to the number of pixels, and it achieves excellent results compared to traditional methods [19,20]. At the same time, another multi-agent algorithm is proposed in Ref. [11], where each agent controls a sliding square window. This method is more robust and has better generalization based on the results of the experiments on the CIFAR-10 dataset and the MNIST dataset [21].

The idea of this work is to use multiple isomorphic agents with sliding square window to find the defect regions. Meanwhile, GRU (Gated Recurrent Unit) [22] models are used to aggregate beliefs on the trajectories of the agents. Compared with the original image, the size of the observation image of each agent has been significantly reduced, and the image category can be judged more quickly. In the following part of this paper, the multi-agent problem is constructed in Section 2. The proposed method and the corresponding models are described in Section 3. In Section 4, experiments are performed on the casting image dataset and compared with several mainstream convolutional neural network based models. The conclusions are given in the last section.

## 2. Multi-Agent Reinforcement Learning

For the problem of casting image classification, a decentralized partially observable Markov decision process model can be used [23,24,25,26,27], which can be expressed as an eight-tuple 〈N,S,A,T,R,O,Z,γ〉, where *N* represents the number of agents, usually greater than or equal to 2 in the case of multi-agent reinforcement learning. *S* represents the state space. $A=A_{1}\times A_{2}\times \cdots \times A_{N}$ represents the joint action space of all agents, where $A_{i}$ represents the action space of the i-th agent. T:S×A×S→[0,1] is the state transition function, which is used to represent the probability of the agent transitioning to state $s^{\prime }\in S$ after performing a joint action a∈A in state s∈S. R:S×A×S→R is the reward function, which represents the reward obtained by the agent after the state transition occurs. Under fully cooperative tasks, the reward value of all agents can be the same. $O=O_{1}\times O_{2}\times \cdots \times O_{n}$ represents the joint local observation space of all agents, where $O_{i}$ represents the local observation space of the i-th agent. Z:S×A→O is the observation function. The i-th agent can only obtain local observations $O_{i}$ under the condition of s∈S. γ∈[0,1] is the discount factor.

According to the different training schemes, multi-agent reinforcement learning methods can be divided into three categories, namely CTCE (Centralized Training Centralized Execution), DTDE (Distributed Training Decentralized Execution), and CTDE (Centralized Training Decentralized Execution) [28,29,30]. In the CTCE scheme, there is a centralized controller. All agents submit their observations and rewards to the centralized controller, and then the centralized controller decides the action for each agent. In the DTDE scheme, each agent interacts with the environment independently and determines actions by itself. In the CTDE scheme, the centralized controller knows each agent’s observations, actions, and rewards during the training phase. After training, the centralized controller is not needed [31,32]. This paper adopts the DTDE scheme, does not require a centralized controller, and adds a communication mechanism, and each agent can obtain information from other agents, as shown in Figure 1.

According to the different types of tasks, multi-agent reinforcement learning methods are divided into three categories, namely the fully competitive, fully cooperative, and mixed task [33,34,35]. In the fully competitive task, assuming there are two agents, the reward value $R_{1}$ of the first agent and the reward value $R_{2}$ of the second agent have the relationship $R_{1}=-R_{2}$. The competitive task emphasizes the performance of the single agent. However in this paper, agents are fully cooperative. In the fully cooperative tasks, the rewards can be jointly maximized since all of the agents’ goals are the same. The joint performance, instead of the performances of the single agents, is emphasized, which makes for a shorter computational time. The common goal of the agents in this paper is to reduce the prediction errors, which will be described in the next section.

## 3. Network Structure and Training Method

The overall network includes five modules, namely the feature extraction module, position encoding module, prediction module, resolution module, and communication module. Taking two agents as an example, the overall network structure is shown in Figure 2. The specific structure of each module will be introduced in the following few sections.

### 3.1. Feature Extraction Module

First, local observations are generated from the original image and the coordinates of the i-th agent, which can be expressed as follows: (1)$$o_{i}\left(t\right)=O\left(I,p_{i}\left(t\right),w\right)$$
where *I* is the original image, $p_{i}\left(t\right)$ is the position coordinate of the i-th agent at time *t*, and *w* is the window size. The initial position coordinate of the i-th agent $p_{i}\left(0\right)$ is randomly generated with uniform probability. By default, the window is set to 20×20 pixels because many experiments have shown that a window of 20×20 pixels can already achieve good results. Taking two agents as an example, the schematic diagram of local observation is shown in Figure 3. As can be seen from Figure 3, the i-th agent move one step to the right and the i+1-th agent move one step down from time *t* to t+1. The step length here is equivalent to the window size.

After the local observation is generated, the feature of the local observation is extracted by the convolutional neural network. Taking the local observation of 20×20 pixels as an example, the structure of the convolutional neural network is shown in Figure 4. It includes three convolutional layers and two max-pooling layers and finally passes through the flatten layer. Using $\theta_{1}$ to represent the parameters in the convolutional neural network, the feature extraction process can be expressed as follows: (2)$$b_{i}\left(t\right)=f_{1}\left(o_{i}\left(t\right);\theta_{1}\right)$$

### 3.2. Position Encoding Module

The position encoding module is the encoding of the agent’s position, which consists of a fully connected layer, and the CeLU (Continuously Differentiable Exponential Linear Units) [36] function is used as the activation function, as shown in Figure 5. The CeLU function is differentiable at all points, and gradient vanishing or gradient exploding problems can be avoided by using CeLU as the activation function. The positions of the agents are quite critical in this method, since it directly determines the performance of the prediction. If the agent is right at the location of the defect, the accuracy of the output can be much higher. The position encoding process can be expressed as follows: (3)$$\lambda_{i}\left(t\right)=f_{2}\left(p_{i}\left(t\right);\theta_{2}\right)$$
where $\theta_{2}$ represents the parameters of the position encoding module. The CeLU function is shown as follows: (4)CeLU(x)=max(0,x)+min(0,exp(x)−1)

### 3.3. Prediction Module

The prediction module consists of the prediction GRU and the classifier. Each agent is assigned a prediction GRU to enable the agent to learn long-term dependencies. The function of this GRU is to aggregate the posterior information throughout the task. The structure of the GRU is shown in Figure 6. The update method of GRU is given by the following four formulas: (5)$$z\left(t\right)=\sigma \left(W_{z}\cdot \left[h\left(t\right),x\left(t\right)\right]\right)$$
(6)$$r\left(t\right)=\sigma \left(W_{r}\cdot \left[h\left(t\right),x\left(t\right)\right]\right)$$
(7)$$\tilde{h}\left(t\right)=\tanh\left(W\cdot \left[r\left(t\right)*h\left(t\right),x\left(t\right)\right]\right)$$
(8)$$h\left(t+1\right)=\left(1-z\left(t\right)\right)*h\left(t\right)+z\left(t\right)*\tilde{h}\left(t\right)$$
where σ represents the Sigmoid function, $W_{z}$, $W_{r}$, and *W* are all parameters that need to be trained.

Let $h_{i}\left(t\right)$ denote the hidden state of the *i*-th agent at time *t*. The update method of prediction GRU is shown as follows: (9)$$h_{i}\left(t+1\right)=f_{3}\left(h_{i}\left(t\right),x_{i}\left(t\right);\theta_{3}\right)$$
where $\theta_{3}$ represents the parameters in the prediction GRU, and $x_{i}\left(t\right)$ consists of three parts: the features of local observations of the i-th agent, the average message from other agents and the position encoding of the i-th agent. The classifier is composed of two fully connected layers, and CeLU is used as the activation function, which can be expressed as follows: (10)$$q_{i}=f_{4}\left(h_{i}\left(T\right);\theta_{4}\right)$$
where *T* represents the time step of each episode, $h_{i}\left(T\right)$ represents the hidden state of the prediction GRU at time *T*, $\theta_{4}$ represents the parameters of the classifier and $q_{i}$ represents the prediction category. The final image prediction is made jointly by all agents, which can be expressed as follows: (11)$$q_{c}=\mathrm{argmax}\left(\mathrm{Softmax}\frac{1}{N}\sum_{i=1}^{N}q_{i}\right)$$

### 3.4. Resolution Module

The resolution module consists of the decision GRU and the policy network. Note that the decisions of the agents not only depend on the current observations, but also on their past observations and states, so the GRU model is suitable for making decisions based on the current and the past information. Each agent is assigned a decision GRU whose structure is the same as the prediction GRU, but with slightly different inputs, which is updated as follows: (12)$${\hat{h}}_{i}\left(t+1\right)=f_{5}\left({\hat{h}}_{i}\left(t\right),x_{i}\left(t\right);\theta_{5}\right)$$
where ${\hat{h}}_{i}\left(t\right)$ is the hidden state of the decision GRU of the *i*-th agent at time *t*, $\theta_{5}$ is the parameter in the decision GRU, and then we input ${\hat{h}}_{i}\left(t+1\right)$ into the policy network to obtain the action at time t+1. The process can be expressed as follows: (13)$$a_{i}\left(t+1\right)=f_{6}\left({\hat{h}}_{i}\left(t+1\right);\theta_{6}\right)$$
where $\theta_{6}$ represents the parameters of the policy network, which is constructed by two fully connected layers, using CeLU as the activation function, and finally passing through the softmax layer, then the position $p_{i}\left(t+1\right)$ at time t+1 is generated from the action $a_{i}\left(t+1\right)$ of the i-th agent at time t+1 and the position $p_{i}\left(t\right)$ at time *t*, which can be expressed as follows:(14)$$p_{i}\left(t+1\right)=g\left(p_{i}\left(t\right),a_{i}\left(t+1\right)\right)$$

### 3.5. Communication Module

Each agent can obtain messages from other agents. The communication module is constructed using CeLU as the activation function by two fully connected layers. The communication process can be expressed as follows: (15)$$m_{i}\left(t\right)=f_{7}\left(h_{i}\left(t\right),{\hat{h}}_{i}\left(t\right);\theta_{7}\right)$$
where $h_{i}\left(t\right)$ and ${\hat{h}}_{i}\left(t\right)$ come from the prediction GRU and decision GRU, respectively, and $\theta_{7}$ represents the parameters of the communication module. So far, $x_{i}\left(t\right)$ can be expanded as follows: (16)$$x_{i}\left(t\right)=\left(b_{i}\left(t\right),{\bar{m}}_{i}\left(t\right),\lambda_{i}\left(t\right)\right)$$
where ${\bar{m}}_{i}\left(t\right)$ is the average message obtained by other agents, which can be given by the following equation: (17)$${\bar{m}}_{i}\left(t\right)=\frac{1}{N-1}\sum_{j\neq i}m_{j}\left(t\right)$$
where $m_{j}\left(t\right)$ represents the message of other agents except the *i*-th agent.

### 3.6. Prediction Process and Training Method

The overall prediction process is shown in Algorithm 1.

In this paper, the stochastic gradient descent method is used to update the network parameters, and $\Theta =\left[\theta_{1},\theta_{2},\theta_{3},\theta_{4},\theta_{5},\theta_{6},\theta_{7}\right]$ represents the parameters defined by Algorithm 1. Let T denote all possible trajectories, and let τ indicate the trajectory sampled from T. Taking the negative of the error between the actual label and the predicted label as the reward value of this trajectory, it can be expressed as follows: (18)$$r_{\tau }=-L\left(q_{c}-\hat{q}\right)$$
where $\hat{q}$ represents the actual label, and the mean squared error loss function is used to replace L. The goal is to maximize the expected reward value *J*, which can be expressed as follows: (19)$$J=E\left(r_{\tau }\right)=\sum_{\tau \in T}p_{\tau }r_{\tau }$$
where $p_{\tau }$ represents the probability of the occurrence of this trajectory, which is expanded as follows: (20)$$p_{\tau }=\prod_{t=0}^{T}\pi \left(a^{t}|s^{t}\right)=\prod_{t=0}^{T}\prod_{i=1}^{N}\pi_{i}\left(a_{i}^{t}|o_{i}^{t}\right)$$
**Algorithm 1:**Multi-agent prediction of image classes
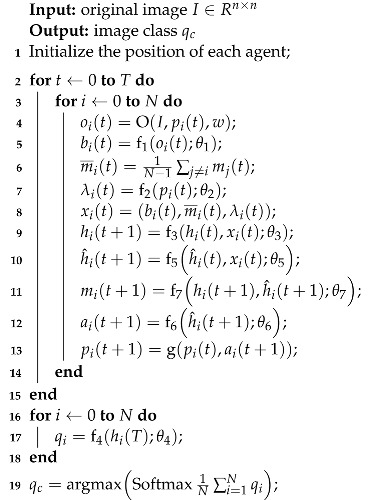


To simplify the formula, denote $a_{i}\left(t\right)$ as $a_{i}^{t}$, and denote $o_{i}\left(t\right)$ as $o_{i}^{t}$. Then we need to calculate the gradient of *J*, which is expanded as follows: (21)$$\begin{array}{cc} \nabla_{\Theta }J & =\sum_{\tau \in T}r_{\tau }\nabla_{\Theta }p_{\tau }+p_{\tau }\nabla_{\Theta }r_{\tau }\\ \; & =\sum_{\tau \in T}p_{\tau }\nabla_{\Theta }\left(\ln p_{\tau }\right)r_{\tau }+p_{\tau }\nabla_{\Theta }r_{\tau }\\ \; & =E\left(\nabla_{\Theta }\left(\ln p_{\tau }\right)r_{\tau }+\nabla_{\Theta }r_{\tau }\right) \end{array}$$
where $r_{\tau }$ is related to the parameter Θ, so the gradient must be calculated for $r_{\tau }$. Because it is not possible to traverse all trajectories, a Monte Carlo method is used to estimate *J* by sampling multiple times. Suppose *C* trajectories are sampled to estimate *J*, then the gradient of the estimated value can be expressed as follows: (22)$$\nabla_{\Theta }\hat{J}=\frac{1}{C}\sum_{k=1}^{C}\nabla_{\Theta }\left(\ln p_{k}\right)r_{k}+\nabla_{\Theta }r_{k}$$
where $\nabla_{\Theta }\hat{J}$ is an unbiased estimate of $\nabla_{\Theta }J$, which is as follows: (23)$$E\left(\nabla_{\Theta }\hat{J}\right)=\nabla_{\Theta }J$$

Therefore, the gradient $\nabla_{\Theta }\hat{J}$ obtained according to Algorithm 1 converges to the actual gradient $\nabla_{\Theta }J$ by probability, so the convergence of the Algorithm 1 is guaranteed.

## 4. Experiment Results and Discussion

### 4.1. Dataset and Setups

In terms of hardware, the machine memory is 16GB, the graphics card is NVIDIA TITAN Xp, the video memory is 12 GB, the CPU is Intel Xeon E5-2680 v4, and the clock speed is 2.4 GHz. In terms of software, the Pytorch 1.8.1 framework is used, the programming language is Python3.8 and the CUDA version is 11.1. The size of each image is 300×300 pixels. The training dataset contains 2875 non-defective casting images and 3758 defective casting images; the test dataset includes 262 non-defective casting images and 453 defective casting images.

In this data set, the casting defects can be mainly divided into two categories: open holes and casting fins. The defects of open holes can be caused by trapped air when the metal is poured into the mold. The defects of the casting fins show up when some extra materials are attached to the edges of the casting. Examples of both type of the defects are shown in Figure 7. However, this paper only studies the surface defects of the castings, which can be detected by visual signals.

There are four choices for each agent’s action, which are up, down, left, and right, each time moving a window size. The position of each agent is represented by two numbers, which represent the coordinates of the agent in the original image. If the agent’s action is unavailable, the position will not change. For example, if the agent’s local observation is already at the far right of the original image at time *t*, but the action at time t+1 is to move to the right, the agent’s position remains unchanged. By default, 10 agents are constructed, the time step of each episode is 5, the local observation of each agent is 20×20 pixels, the number of trajectory sampling is three times and the learning rate is 0.002. The same preprocessing was performed on the dataset for each experiment. Compare our method with GhostNet [37], MobileNetV3 [38], Res2Net [39], and ShuffleNetV2 [40]. This paper uses the F1-score as the performance indicator. The F1-score is the harmonic mean of precision and recall, which can be described as follows: (24)$$F1=2\frac{precison*recall}{precison+recall}$$

### 4.2. Performance Analysis under Different Parameters

Under the default settings, the algorithm is run for 200 epochs, as shown in Figure 8. As can be seen from Table 1, the top-1 F1-score in the test dataset can reach 0.95551, and the prediction time of each epoch is only 1.84 s. To better evaluate the proposed algorithm, a comparative experiment is designed for four parameters: the time step *T* of each episode, the number of agents *N*, the number of trajectory sampling times *C* and the window size *w*. Changes are made each time based on the default settings.

To better show the moving process of agents, taking T=3 as an example, the moving process of 10 agents is shown in Figure 9. It can be seen that the two agents on the far left did not move in the second step because, at this time, the agent’s action was to move to the left, but it was unable to move to the left.

First, we verify the impact of the time step *T* of each episode on the performance of the algorithm, and modify the *T* value based on the default setting, as shown in Figure 10. It can be seen that as the *T* value increases, the algorithm converges faster. As can be seen from Table 1, with the *T* value increases, the top-1 F1-score also increases. When T=9, the average training time per epoch increases by 147% compared to when T=3, but the average test time per epoch only rises by 58%. The change in *T* value is more of an impact on the training time.

By modifying the number *N* of agents based on the default settings, the change in the F1-score is shown in Figure 11. It can be seen that when the number of agents is increased to 15, the curve of the F1-score changes significantly and can converge faster. As can be seen from Table 1, for each additional five agents, the average training time per epoch will increase by about 15%, and the average test time per epoch will increase by about 10%. When the number of agents is 20, the top-1 F1-score in the test dataset can reach 0.98677. Compared with the case of N=5, the number of agents has become four times the original, but the average training time per epoch only increases by 47.7%, and the average test time per epoch increased by 33%.

Modifying the number of trajectory samples *C* based on the default settings, the change in the F1-score is shown in Figure 12. As can be seen from the figure, when C=9, the F1-score of the training dataset is relatively stable in the later stage. It can be seen from Table 1 that the change of the *C* value mainly affects the training time, and has little effect on the test time.

Modifying the window size *w* based on the default settings, the change in the F1-score is shown in Figure 13. As can be seen from Figure 13, when w = 10, the F1-score of the training dataset is lower, and the convergence speed is also accelerated as the window size increases. It can be seen from Table 1 that the change of window size has less impact on the training and test time, but when the window size becomes too small, the F1-score of the test dataset drops a lot.

In Figure 14, the result of a non-defective case and two defective cases are given. In the left sub-figure, no agent has detected any defects. In the middle sub-figure of Figure 14, a casting fin defect is detected by the agent, whose observation window is marked in green. In the right sub-figure of Figure 14, a open hole defect is also detected by the agent, whose observation window is marked in green. All the observation windows of the normal agents are marked in red.

### 4.3. Comparative Experiments

The GhostNet, MobileNetV3, Res2Net, and ShuffleNetV2 models are selected for comparative experiments. All four networks use the cross-entropy loss function, the learning rate is set to 0.0003, the batch size is set to 32, and each model is trained for 50 epochs. The F1-scores of each network are shown in Figure 15.

The top-1 F1-score of each network and the average time of per epoch are shown in Table 2. It can be seen from the table that the average test time of each epoch of the comparison model is at least 10.90 s, while the algorithm proposed in this paper only needs 1.84 s under the default setting. When the number of agents is 20, the top-1 F1-score of this algorithm in the test dataset reaches 0.98677, which is only 0.00754 lower than GhostNet, but the average test running time per epoch is only 20.55% of GhostNet.

## 5. Conclusions and Future Works

This paper uses multiple agents to jointly predict the image category, significantly reducing the prediction time and the total calculation amount. This method can be applied to devices with weak computing power, since the input of each device is only a tiny part of the original image. The joint operation of multiple agents not only ensures accuracy but also reduces the computational burden of each device. With the increasing size of images today, a 2 K or even 4 K image requires considerable computing resources on traditional convolutional neural networks for classification, so the method in this paper has great potential for large-size image classification. For the casting dataset, compared with the traditional convolutional neural network GhostNet, the prediction time of the method in this paper is only one-fifth of the original, which has made significant progress in saving computing resources and reducing the defects detection time.

The proposed method in this work can be extended in several aspects in future works. First, the way of communication between the agents can be further investigated, by which the observation of each single agent can be better utilized. Second, instead of being a fixed value, the step size can be determined in an adaptive manner in the future methods. Third, the sizes or even the shapes of the observation windows can also be changed in the process with respect to the agents’ states. All these possible future works can make further improvements on the prediction accuracy of the casting defects detection or the corresponding computational burdens.

## Figures and Tables

**Figure 1 sensors-22-05143-f001:**
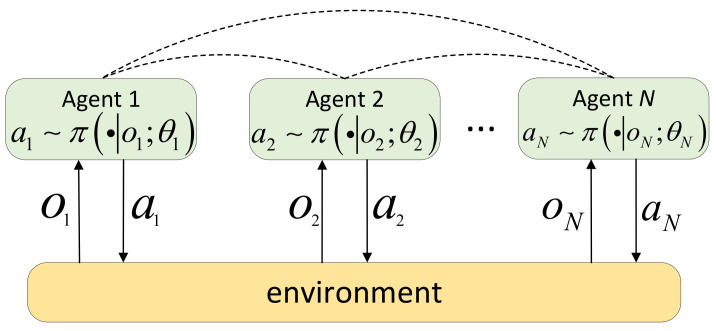
Schematic diagram of distributed training decentralized execution.

**Figure 2 sensors-22-05143-f002:**
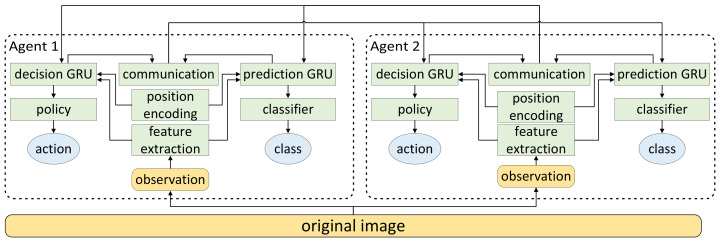
The overall network structure.

**Figure 3 sensors-22-05143-f003:**
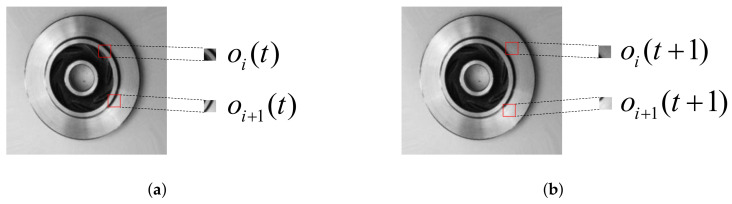
Schematic diagram of local observations. (**a**) The local observations of the i-th agent and the i+1-th agent at time *t*; (**b**) The local observations of the i-th agent and the i+1-th agent at time t+1.

**Figure 4 sensors-22-05143-f004:**
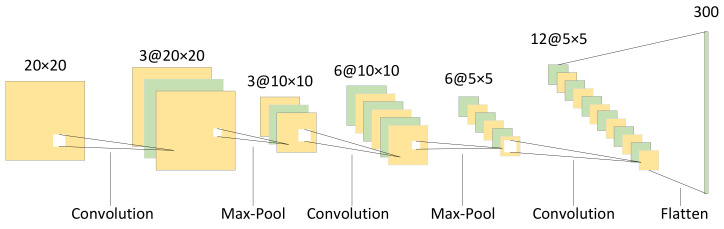
The structure of the convolutional neural network.

**Figure 5 sensors-22-05143-f005:**
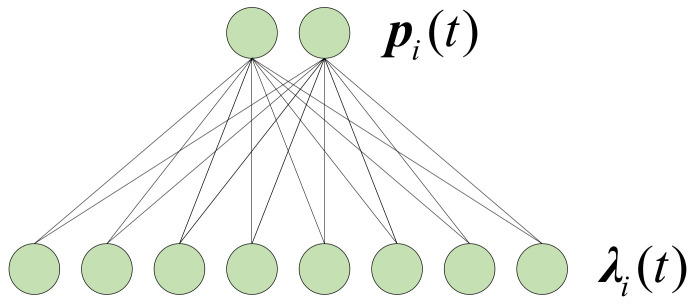
The position encoding module.

**Figure 6 sensors-22-05143-f006:**
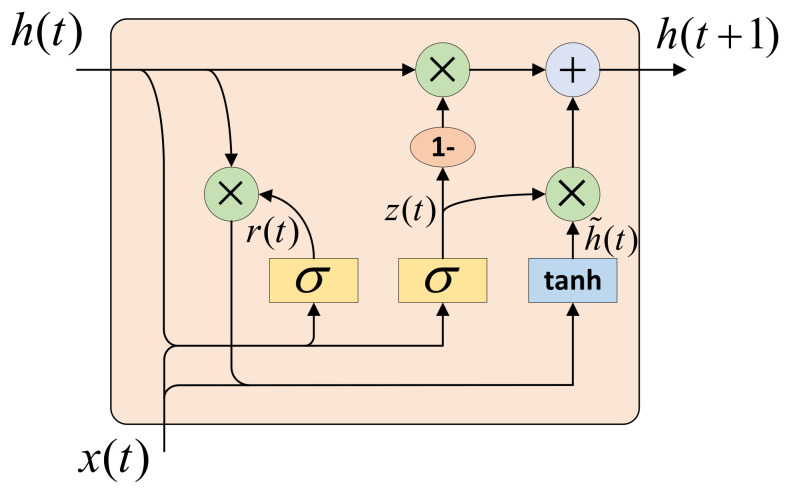
The structure of GRU.

**Figure 7 sensors-22-05143-f007:**
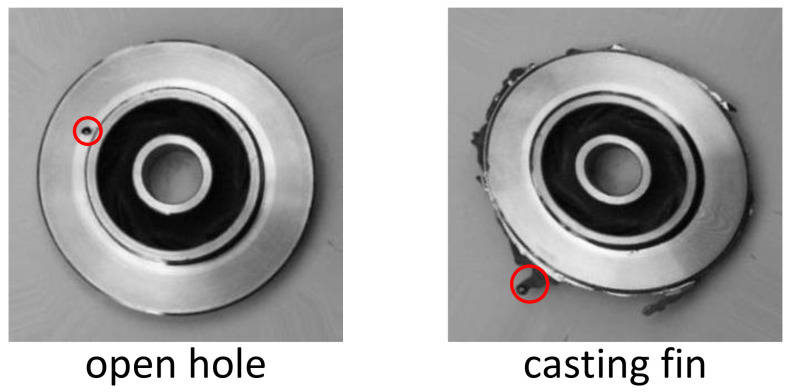
The examples of the open hole defect and the casting fin defect. The red circles are manually marked.

**Figure 8 sensors-22-05143-f008:**
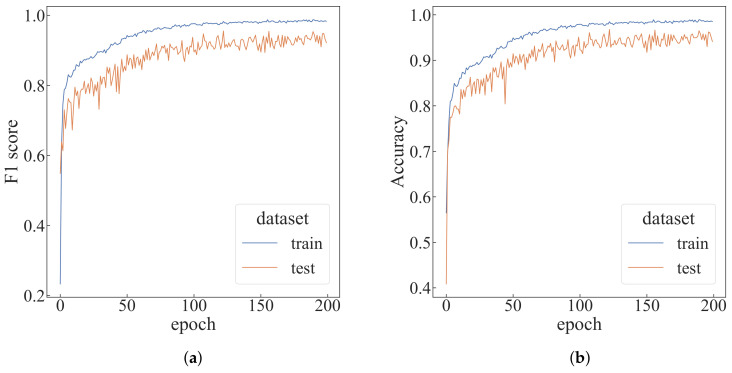
The performance of default settings. (**a**) F1-score of default settings; (**b**) accuracy of the default settings.

**Figure 9 sensors-22-05143-f009:**
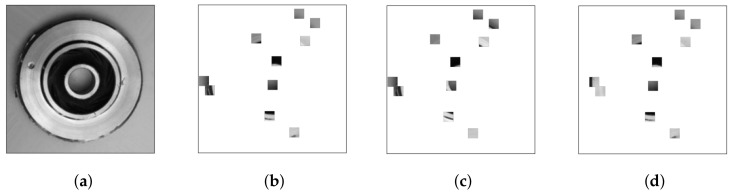
The moving process of agents. (**a**) original image; (**b**) step = 1; (**c**) step = 2; (**d**) step = 3.

**Figure 10 sensors-22-05143-f010:**
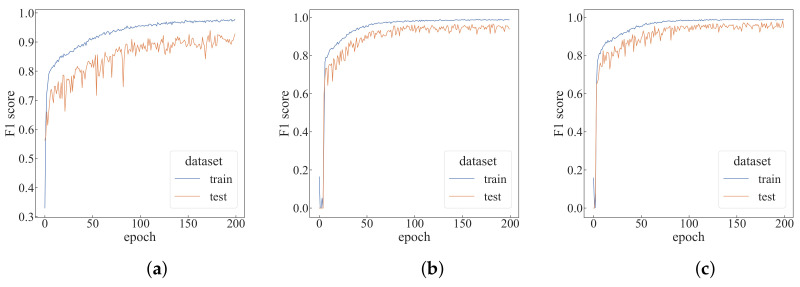
Performance comparison with different time steps. (**a**) T=3; (**b**) T=7; (**c**) T=9.

**Figure 11 sensors-22-05143-f011:**
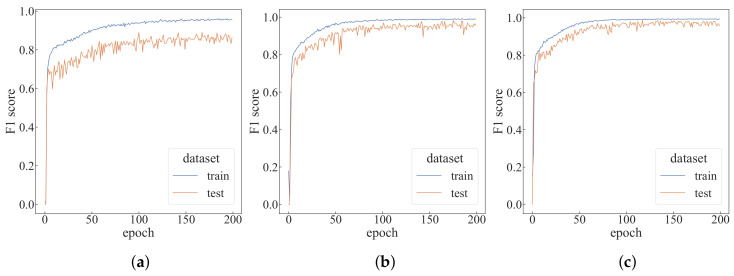
Performance comparison with different numbers of agents. (**a**) N=5; (**b**) N=15; (**c**) N=20.

**Figure 12 sensors-22-05143-f012:**
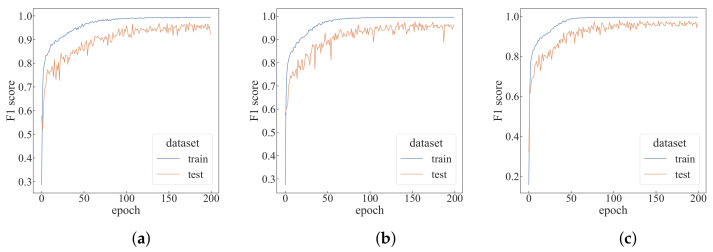
Performance comparison with different trajectory sampling times. (**a**) C=5; (**b**) C=7; (**c**) C=9.

**Figure 13 sensors-22-05143-f013:**
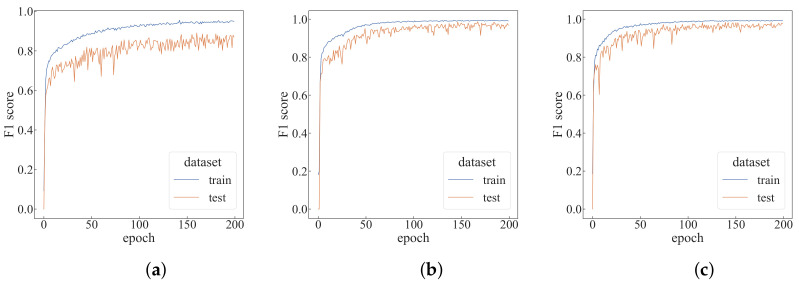
Performance comparison with different window sizes. (**a**) w=10; (**b**) w=30; (**c**) w=40.

**Figure 14 sensors-22-05143-f014:**
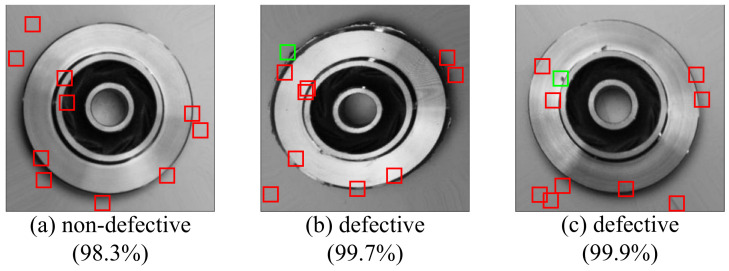
The prediction results of a defective casting and two non-defective castings. (**a**) A non-defective case, whose probability of being non-defective is 98.3%; (**b**) A defective casting case (with fins), whose probability of being defective is 99.7%; (**c**) A defective casting case (with open holes), whose probability of being defective is 99.9%. The observation windows of the normal agents are marked in red, and the observation windows of the defects detected agents are marked in green.

**Figure 15 sensors-22-05143-f015:**
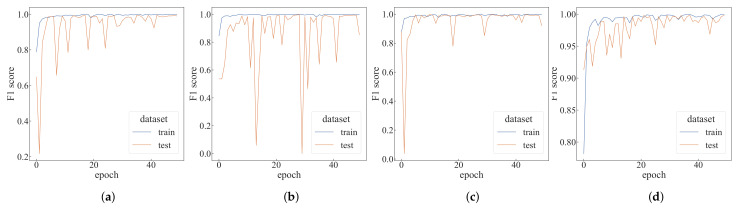
The F1-score of different models. (**a**) GhostNet 1.0×; (**b**) MobileNetV3 Small; (**c**) Res2Net-50; (**d**) ShuffleNetV2 1.0×.

**Table 1 sensors-22-05143-t001:** Performance comparison with different parameters.

Parameters	Top-1 Training F1-Score	Top-1 Test F1-Score	Average Training Time Per Epoch (s)	Average Test Time Per Epoch (s)
N=5, T=5, w=20, C=3	0.96077	0.89051	39.58	1.68
N=10, T=3, w=20, C=3	0.97834	0.93985	30.62	1.67
N=10, T=5, w=10, C=3	0.95514	0.89051	44.80	1.84
N=10, T=5, w=20, C=3	0.98789	0.95551	44.85	1.84
N=10, T=5, w=20, C=5	0.99515	0.97164	67.28	1.85
N=10, T=5, w=20, C=7	0.99653	0.97692	94.18	1.83
N=10, T=5, w=20, C=9	0.99688	0.98099	117.51	1.86
N=10, T=5, w=30, C=3	0.99463	0.98667	44.89	1.85
N=10, T=5, w=40, C=3	0.99446	0.98305	46.25	1.88
N=10, T=7, w=20, C=3	0.99084	0.96629	59.79	2.41
N=10, T=9, w=20, C=3	0.99084	0.98299	75.76	2.64
N=15, T=5, w=20, C=3	0.99153	0.98292	52.58	2.05
N=20, T=5, w=20, C=3	0.99498	0.98677	58.46	2.24

**Table 2 sensors-22-05143-t002:** Performance comparison of the different models.

Models	Top-1 Training *F*1-Score	Top-1 Test *F*1-Score	Average Training Time Per Epoch (s)	Average Test Time Per Epoch (s)
GhostNet 1.0×	1	0.99431	110.93	10.90
MobileNetV3 Small	1	0.99809	129.54	13.59
Res2Net-50	0.99965	0.99809	185.67	12.81
ShuffleNetV2 1.0×	0.99983	0.99809	126.63	13.64

## Data Availability

Not applicable.

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
