# Peer review of "Image Classification Method Based on Multi-Agent Reinforcement Learning for Defects Detection for Casting"

_sensors, 2022, doi:10.3390/s22145143_

Round 1

Reviewer 1 Report

The submitted paper (sensors-1798726) entitled: “Casting Image Classification Method Based on Multi-Agent Reinforcement Learning for Defects Detection” proposes a casting image classification method based on multi-agent reinforcement learning. The aim was to reduce the detection time and the total calculation amount using multiple agents to observe only a small part of the image and move freely on the image to judge the result together.

·       The meaning of the following terms could be explained the first time they appeared in the manuscript, especially for the readers who are not familiar: CeLU and GRU.

·       The figures are not clear enough, especially Figs. 1 and 2.

·  Although the new method proposed has noticeable novelty, the presentation lacks commentary and information that could be helpful to the readers of the paper and promote the presented research.

·       Abstract and conclusion are briefly described and inadequately verified, lacking scientific value.

·       A few details concerning the Schematic diagram of local observation in Fig. 3 could be added especially for the readers who are not familiar.

  • The Authors are invited to justify the selection of the specific experiments for analysing.

Reviewer 2 Report

Study on the deep learning method to deal with the defects detection is a new method. In this paper an image classification method based on multi-agent reinforcement learning is proposed to solve the problem of casting defects detection.

The titile is better to revised as "Image Classification Method Based on Multi-Agent Reinforcement Learning for Defects Detection for Casting".

what kind of defects? their size, geometry and color?

How to determine the initial postions of agents?

The lines and arrows involved h(t)and x(t) in Fig.6 are not proper.

The multi-agent method was validated to reduce the computing time. But, there is no results for the acutal prediction. Please provide the prediciton results for some cases, and give examples of the training, test cases. 
